# Integrating Artificial Intelligence for Advancing Multiple-Cancer Early Detection via Serum Biomarkers: A Narrative Review

**DOI:** 10.3390/cancers16050862

**Published:** 2024-02-21

**Authors:** Hsin-Yao Wang, Wan-Ying Lin, Chenfei Zhou, Zih-Ang Yang, Sriram Kalpana, Michael S. Lebowitz

**Affiliations:** 1Department of Laboratory Medicine, Linkou Chang Gung Memorial Hospital, Taoyuan 33343, Taiwan; mdhsinyaowang@gmail.com (H.-Y.W.); wal018@health.ucsd.edu (W.-Y.L.);; 2School of Medicine, National Tsing Hua University, Hsinchu 300044, Taiwan; 320/20 GeneSystems, Gaithersburg, MD 20877, USA

**Keywords:** AI, multi-cancer early detection (MCED), serum biomarkers

## Abstract

**Simple Summary:**

Governments worldwide have prioritized multicancer early detection (MCED) for the better management of cancers. Artificial intelligence (AI) is a promising technology to enhance the performance of MCED. In this review, key components of MCED AI are explored. We focus on detection targets such as serum protein biomarkers and cell-free DNA. Based on the serum biomarkers, various AI model training methods and validation techniques are investigated. The emphasis is on understanding how these approaches influence predictive efficacy. We demonstrate the importance of real-world data rather than case-control data for trustworthy implementation and the potential benefits of AI integration in MCED. Moreover, challenges in deploying MCED AIs in clinical settings are highlighted, including issues such as presenting predictive reports and addressing cancer-related information.

**Abstract:**

The concept and policies of multicancer early detection (MCED) have gained significant attention from governments worldwide in recent years. In the era of burgeoning artificial intelligence (AI) technology, the integration of MCED with AI has become a prevailing trend, giving rise to a plethora of MCED AI products. However, due to the heterogeneity of both the detection targets and the AI technologies, the overall diversity of MCED AI products remains considerable. The types of detection targets encompass protein biomarkers, cell-free DNA, or combinations of these biomarkers. In the development of AI models, different model training approaches are employed, including datasets of case-control studies or real-world cancer screening datasets. Various validation techniques, such as cross-validation, location-wise validation, and time-wise validation, are used. All of the factors show significant impacts on the predictive efficacy of MCED AIs. After the completion of AI model development, deploying the MCED AIs in clinical practice presents numerous challenges, including presenting the predictive reports, identifying the potential locations and types of tumors, and addressing cancer-related information, such as clinical follow-up and treatment. This study reviews several mature MCED AI products currently available in the market, detecting their composing factors from serum biomarker detection, MCED AI training/validation, and the clinical application. This review illuminates the challenges encountered by existing MCED AI products across these stages, offering insights into the continued development and obstacles within the field of MCED AI.

## 1. Introduction

### 1.1. Background and Motivation

Historically, cancer early detection has revolved around procedures like the Pap test introduced around 100 years ago [1]. Early cancer detection is a pivotal factor in improving patient survival rates, as it allows for earlier intervention, including the surgical removal of localized solid tumors, thus avoiding the critical metastatic stage where survival rates sharply decline to less than 50%, even with the most advanced systemic therapies [2]. In many instances, cancer progresses over the course of years from its initial site to the point of metastasis, presenting a window of opportunity for early detection [3]. The COVID-19 pandemic disrupted healthcare systems globally, leading to the postponement or cancelation of nonessential medical services, including cancer screenings. Recent increases in cancer-related mortality rates observed might partly be attributed to delays in cancer detection due to disrupted screening services [4].

Early cancer detection would improve prognosis; however, this conventional approach faces notable challenges. In fact, a substantial proportion (~57%) of cancer-related deaths in the United States are attributed to cancers that currently lack effective screening protocols [5]. For decades, cancer early detection has focused on the detection of individual cancer types. These methods include low-dose chest computed tomography (CT) for lung cancer, mammography for breast cancer, Pap smear for cervical cancer, and fecal occult blood tests for colorectal cancer [6]. Although these screening methods have contributed significantly to the early detection of cancer, the cancer types that the methods screen represent only a fraction of the overall cancer landscape [7]. Only 14% of cancer cases in the US are diagnosed through recommended screening tests, highlighting the limitations of traditional screening methods. The majority of cancer diagnoses occur after symptoms have manifested or during unrelated medical visits [8]. Moreover, cancer screening by the individual cancer-type manner is inconvenient and individuals may find themselves required to visit multiple healthcare providers to undergo various screening tests. These drawbacks contribute to a reduced adherence to cancer screening through these methods [9].

The inadequacy of current cancer screening strategies can be attributed to several factors. These include (1) suboptimal adherence to screening guidelines, (2) disparities in access to screening, (3) limitations in existing screening technologies, (4) and the occurrence of cancers between recommended testing windows [10]. Consequently, a significant portion of cancer cases in the United States are detected at advanced stages, where treatment becomes substantially more challenging. For instance, among cancers with established screening protocols, the proportion of late-stage diagnoses ranges from 20.9% for prostate cancer to a staggering 64.7% for lung cancer.

This introduction sets the foundation for exploring the promise of multicancer early detection (MCED) strategies, which aim to revolutionize cancer screening by addressing these challenges. Subsequent sections will delve into advances in MCED approaches and the pivotal role played by artificial intelligence (AI) in enhancing the early detection of a broad spectrum of cancers, ultimately improving survival rates and treatment outcomes.

### 1.2. Role of Artificial Intelligence in MCED

MCED involves detecting different cancer types. Some of the cancer types share the same molecular behaviors, while some of them do not. To address a complicated task like MCED, massive data would be necessary to provide sufficient information for good discrimination. In analyzing massive data, AI is good at detecting hidden patterns within complicated datasets. Typically, supervised machine-learning algorithms are the workable AI approach in the biomedical field. Because of its outstanding performance in classification or prediction, AI has been widely applied across multiple biomedical fields in recent years. The applications include correlating genetic data to obesity [11], liquid biopsies for predicting cancer metastasis [12], clinicopathological data for the risk stratification of cancer [13], genetic data for MCED [14], and protein biomarkers for MCED [15].

In the field of MCED, applying AI has also proven to be efficient in improving diagnostic performance [16]. Harnessing AI has become a must-use technology in analyzing MCED data because MCED tools typically target tens of analytical targets. Interpretation of the complicated patterns that are composed of tens of components would be difficult without AI approaches. Even when some biomarker experts are good at interpreting complicated data patterns, the interpretation job is still time-consuming and labor-intensive. The integration of AI algorithms can significantly enhance the MCED tools of liquid biopsy or genetic data analysis because the data generated in the tests are massive, complicated, and possibly contain multiomic data. AI facilitates multimodal analysis by integrating genomic, proteomic, and metabolomic data from liquid biopsies, providing a holistic view for the improved accuracy of MCED. In the genetic data analysis of MCED, the number of analytic targets is typically large. To improve the analysis of the massive data, AI excels in recognizing intricate patterns and associations, offering valuable insights into cancer detection, mutation profiles, and hereditary factors. Besides binding multiomic data, AI is also good at the seamless integration of analytical data with clinical information as the clinicopathological data, providing more comprehensive health profiles. Moreover, MCED powered by AI can learn and become updated by new data. The continuous learning of AI would mean the continuous improvement of MCED. While the specific patterns of MCED are learned by AI, the learned knowledge of the AI models would be describable and comparable. In short, applying AI in MCED would render MCED more accurate and objective. The concept of MCED AI is illustrated in Figure 1.

However, the diagnostic performance of using AI in MCED is still discrepant between different studies. There are many possible reasons for the inconsistent results. One significant factor is the variation in study designs. Most related studies use data from case-control studies for the training and validation of AI models [15,17]. Only a few research labs use data from cancer screening for the development and validation of related AI models [6,18]. Using real-world cancer screening data can be quite important because it is the only way to align with the real-world application of these models. AI models are data-driven and would be heavily influenced by the composition of training data. If the data used for training the model differ from the data of the intended-use scenarios, then the use of AI models in real-world healthcare would be significantly limited. Analytical variation between different ethnic groups is also another factor that must be considered. Only when the factors that affect the AI, such as the input analytical data and model training/validation, are well-optimized and standardized can the clinical field fully leverage the capabilities and advantages of AI in cancer screening.

## 2. Trajectory of Early Cancer Detection Methods

### 2.1. Evolutionary Overview of Multiple-Cancer Early Detection

In recent years, significant progress has been made in the realm of cancer screening, especially in the domain of cancer early detection [19]. Several single-cancer screening tools, such as the low-dose chest CT, mammography, Pap smear, and colonoscopy, are currently applied for individual cancers. The evolution of MCED has been driven by the recognition of the limitations inherent in traditional single-screening approaches [19]. In contrast to multimodal single-cancer screening methods, MCED tests aggregate the prevalence of various cancer types within a given population [20]. This approach provides a single, all-encompassing evaluation while maintaining a relatively low false-positive rate [20]. Moreover, the challenge of varying levels of adherence to current screening protocols has complicated cancer detection [21], necessitating the development of noninvasive multicancer screening methods to reduce the morbidity and mortality from cancers.

Historically, the exploration of whole-body imaging and endoscopic techniques has been considered a route to achieving universal cancer screening [19]. Nevertheless, persistent issues, such as high false-positive rates [22] and potential complications arising from radiation exposure [22,23] or invasive procedures [23], still require resolution [24]. In recent times, a revolutionary breakthrough has emerged in the form of liquid biopsies, which analyze cancer-related biomarkers present in body fluids [24]. This development has introduced a transformative dimension to the field, offering a less invasive and more accessible means of early cancer detection. Additionally, AI has been recruited to analyze large amounts of data, including medical images and genetic data, to identify patterns and anomalies indicative of cancer [24]. The incorporation of AI into the screening process increases the accuracy and efficiency of cancer detection. Of note in the current clinical workflow, MCEDs can serve as a precursor to more specific cancer diagnoses. Specifically, MCEDs should not be viewed as diagnostic, but rather as predictive for risk, and these approaches inherently alter the acceptable level of specificity.

### 2.2. Advancements in Imaging and Endoscopic Tools

Imaging techniques, such as CT, magnetic resonance imaging (MRI), and positron emission tomography (PET), have potential in MCED by providing noninvasive methods to identify tumors in asymptomatic patients. However, these techniques have limitations and challenges. Imaging will always be limited to a minimal size of a lesion that is noticeable (even by AI) on a scan. False positives occur when imaging identifies benign abnormalities as suspicious, leading to unnecessary tests, interventions, and mental stress for the patient [22,25]. For example, previous studies about whole-body MRI showed that abnormal findings were expected in about 95% of screened subjects; about 30% of subjects would require further investigations, but less than 2% would be reported as suspicious for malignant cancers [22]. Moreover, there is a carcinogenic risk associated with radiation exposure from these examinations [26]. Annual CT scans from ages 45 to 75 years could result in an increased risk of cancer mortality of 1.9%, or approximately 1 in 50 people [23].

Endoscopic techniques, on the other hand, allow the direct visualization and biopsy of suspicious lesions. The development of novel endoscopic technologies, such as narrow-band imaging (NBI) and confocal laser endomicroscopy (CLE), has enhanced the ability to detect lesions in the gastrointestinal tract and other organs [27]. However, these invasive procedures lack cost-effectiveness and may carry risks such as bleeding and bowel perforation. While the reported post colonoscopy perforation rate is less than 0.1%, it remains a significant concern due to its status as a severe complication associated with high mortality rates [28].

### 2.3. Emergence of Liquid Biopsy-Based Approaches

Liquid biopsy has emerged as a revolutionary technique for MCED. This innovative method through phlebotomy greatly reduces the possible harm associated with more invasive screening methods. Liquid biopsy involves the analysis of disease-related markers found in bodily fluids, encompassing a diverse range of analytes, such as circulating tumor DNA (ctDNA), circulating tumor RNA (ctRNA), circulating tumor cells (CTCs), proteins, and metabolites [24].

Over the past decade, there has been a rapid development and adoption of next-generation-sequencing (NGS)-based methods in cancer research [29]. These methods allowed us to capture tumor-specific genomic aberrations in circulation [29]. There are two primary sources of tumor DNA that can be noninvasively assessed within the circulatory system: ctDNA and CTCs [30]. CtDNA consists of small nucleic acid fragments shed from necrotic or apoptotic tumor cells [31]. In contrast, CTCs represent intact and often viable cells, which may originate from active cell invasion or the passive shedding of tumor cell clusters [30]. Genomic biomarkers hold the potential to provide a more representative ‘summary’ of tumor heterogeneity within a patient, and also open up the possibility of detecting cancer at an early stage [29]. Several commercial products, such as GRAIL, have demonstrated impressive performance in detecting multiple cancer types and identifying their origin within asymptomatic patients [32].

On the other hand, serum protein tumor markers, like CEA, AFP, CA-125, CA-19.9, PSA, and others, have been used for decades to aid in diagnosing and managing various cancers [33]. However, due to their relatively low sensitivity and specificity for early cancer detection, most international guidelines recommend their use primarily for monitoring cancer recurrence or assessing therapy response rather than as screening tools for early detection [6]. One potential strategy to address this limitation involves combining multiple serum markers into diagnostic biomarker panels [15,33,34]. Previous research has shown that, when AI algorithms are employed to train these serum marker panels, the resulting algorithms become effective tools for cancer screening [16]. These AI algorithms consistently exhibit high levels of accuracy, generalizability, and cost-effectiveness, making them promising candidates for improving early cancer detection [6,35].

While imaging or endoscopic tools can be accurate and would provide additional treatments besides diagnosis, some drawbacks, including being labor-intensive, high technique-requiring, and the risks of complications, still hamper the tools for MCED at a large scale. By contrast, liquid biopsy-based approaches hold promise in dealing with the aforementioned drawbacks so as to reach the screening purpose. To depict the pictures of different cancer types, the following section will introduce various serum biomarkers to realize MCED through liquid biopsy-based approaches.

## 3. Serum Biomarkers as Critical Indicators

### 3.1. Protein Biomarkers: Unveiling Diagnostic Potential

Cancer cells or other cell types in the tumor microenvironment release soluble molecules that are identified as serum tumor markers by noninvasive diagnostic assays. These molecules ideally detect disease early, predict the response, and aid in monitoring therapies. For example, in breast cancer, different serum markers are carcinoembryonic antigen (CEA), the soluble form of the MUC-1 protein (CA15-3), circulating cytokeratins, such as tissue polypeptide antigen (TPA), tissue polypeptide-specific antigen (TPS) and cytokeratin 19 fragment (CYFRA 21-1), and the proteolytically cleaved ectodomain of the human epidermal growth factor receptor 2 (s-HER2). These markers are used majorly in follow-up [12], but are not used in screening breast cancer [36].

Protein tumor markers have not been fully exploited clinically both diagnostically and prognostically. Therefore, the expansion from individual protein biomarker analysis to protein panels or proteomes develops a comprehensive prognostic analysis to predict disease onset and progression [37,38]. The protein panel analysis far exceeds the single-biomarker analysis in facilitating specific intervention or guiding treatment, especially in drug resistance. Challenges prevail in the transition from single biomarkers to proteomic panels, both on the basis of process development and technicality. However, recent advancements in the proteomic techniques have fortified that analysis of multiple proteins simultaneously in the blood, urine, cerebrospinal fluid, or any other biological sample [38].

The technical difficulties in tumor marker measurement include errors due to the difference between labs and also within batches. These variation combinations to form a panel result in low robustness and reproducibility. Hence, in the development of a robust panel assay over time and across laboratories, a single analytical parameter determined by a single method permits the quantification of errors and batch variability. Further, results are compared by absolute quantitative technologies rather than relative quantitative techniques. Absolute quantification requires the lack of dependency on affinity reagents, which are instead directed by mass spectrometry-based proteomics [39]. The US FDA has approved 15 protein biomarker assays in serum and/or plasma. Of the 15 FDA-approved protein biomarkers for cancer proteins, 9 are applicable for serum and 6 for serum/plasma. Although both plasma and serum are identical in protein composition, the expression or recovery of individual proteins vary greatly. For instance, the free PSA concentration differs in serum and plasma [40]. The HUman Proteome Organization recommends plasma for proteomics studies [41].

The idea of panel testing for proteomic profiling has emerged as an effective method in the diagnostics of cancer; particularly, cancer proteomics is clinically feasible. The enzyme-linked immunosorbent assay, immunohistochemistry, and flow cytometry system are reliable, sensitive, and widely used in the clinical diagnosis, prognosis, and treatment monitoring of cancer [42]. Alternative techniques, like mass spectrometry, protein arrays, and microfluidics, are extensively used and are being developed for clinical application [43]. On top of the massive data created by panel testing, proteomic workflows for the targeted analysis of protein panels have improved with highly standardized sample-preparation protocols [44], data-independent acquisition techniques [38], sensitivity, and faster mass spectrometers conjoined with micro- and analytical flow rate chromatography [45]. The absolute quantification has improved the statistical analysis, cross-study, and cross-laboratory comparability, simplifying the accreditation of analytical tests [46].

In 2009, OVA1 was approved for the evaluation of ovarian tumors in combination with the measurement of five serum proteins: apolipoprotein A1, β-2 microglobulin, CA -125, transferrin, and transthyretin [47]. In 2011, ROMA was approved for the prediction of ovarian malignancy along with two proteins—human epididymis protein 4 and CA-125 [48]. For the early detection of cancer, a total of 1261 proteins were identified that were involved in oncogenesis, in tumor angiogenesis, differentiation, proliferation, and apoptosis, in the cell cycle, and in signaling. In as many as 1261 proteins, 9 protein biomarkers have been approved as “tumor-associated antigens” by the USFDA. Although these protein biomarkers have not yet been approved for MCED, in many Asian regions, such as China [35], Taiwan [33], the Republic of Korea [49], etc., the use of protein biomarkers for MCED has been put into practice for more than 10 years. The popularity of this approach lies in its convenience, as cancer screening for many different cancer types can be conducted with a simple blood test. This includes many cancer types for which there is no preferred screening method [18]. Additionally, the cost of protein tumor marker tests is relatively low; the cost of one marker test may be around USD 10 or even lower, making it financially feasible for widespread use. In terms of the diagnostic performance, using protein biomarker panels can achieve approximately 40% sensitivity and 90% specificity [33]. In regions with a high accessibility of follow-up diagnostic approaches (e.g., endoscope, CT, and MRI), this is a convenient and competitive approach. The diagnostic performance of the protein biomarkers is summarized in the Appendix A.

In the post human genome project era, the cost of detecting genes or even genomics has kept dropping, making genetic testing approachable and offering promising biomarkers, like protein biomarkers, for MCED. Additionally, genetic biomarkers provide the possibility to detect cancer-driving mechanisms. Testing genes as the biomarkers for MCED will be addressed in the following section.

### 3.2. Cell-Free DNA Biomarkers: Unleashing Genomic Clues

Cell-free DNA (cfDNA) are noninvasive markers detected in serum, plasma, urine, and CSF [50], and a more favored biomarker for cancer, surpassing the gold-standard approach of biopsy sampling, which is invasive with a restricted frequency of usage and site. It depicts tumor heterogeneity with a comprehensive representation, allowing multiple samplings from a single blood draw and represents various tumor clones and sites, providing a comprehensive representation [51]. All cells release cfDNA that may be necrotic or apoptotic. The cfDNA reveals mutations, methylation, and copy number variations that may be related to cancer [52]. Hence, its molecular profiling has a potential role in noninvasive cancer management with the advent of ultrasensitive technologies (e.g., NGS, BEAMing (beads, emulsions, amplification, and magnetics), and droplet digital PCR (ddPCR)). It has evolved as a considerable surrogate marker in tumor detection, staging, prognosis, localization, and in the identification of acquired drug resistance mechanisms [53].

The sensitivity to detect tumor-derived cfDNA is expressed in terms of the mutant allele fraction (MAF), which is the ratio between the amounts of mutant alleles and wild-type alleles in a sample. The MAF detection limits of quantitative PCR ranges between 10 and 20%. However, variations in PCR techniques, like allele-specific amplification [54], allele-specific nonextendable primer blocker PCR [55], and peptide nucleic acid-locked nucleic acid PCR clamp [56], increase the sensitivity. Several genome-wide sequencing methods have been developed in the last decade. The methods include plasma-Seq [57], Parallel Analysis of RNA Ends sequencing [58], and modified fast aneuploidy screening test-sequencing [59] for cfDNA detection at 5–10% MAF. Targeted sequencing approaches include the exome sequencing [60], CAncer Personalized Profiling by deep Sequencing (CAPP-Seq) [61], and digital sequencing [62]. Targeted sequencing approaches are of high coverage, whereas whole-genome sequencing (WGS) approaches are of low coverage. Targeted approaches detect mutations even at a low ctDNA, whereas WGS assess copy number alteration in ctDNA. A lower MAF is obtained with the digital PCR (dPCR) method, including microfluidic-based ddPCR and BEAMing [63] quantified with extreme sensitivity (0.001–0.05% MAF). The multiplexing capabilities are limited, as the primers or probes target specific mutations or loci.

For the purpose of MCED, cfDNA detects a tumor at an asymptomatic stage with a diameter of 5 mm. The ratio of tumor-derived cfDNA to normal cfDNA < 1–100,000 copies (MAF of 0.001%) corresponds to a tumor of 5 mm in diameter [64]. In blood, 1 mL of plasma contains approximately 3000 whole-genome equivalents [65], and in the total 3 L, plasma represents 9,000,000 copies. In the entire cfDNA population, only one cancer genome originates from a 1 mm diameter tumor, increasing the probability of extracting one tumor-derived cfDNA fragment from a 10 mL blood sample, which is very low. Hence, these available methods detect tumors with a diameter greater than 1 cm (0.5 cm^3^) [64]. Different from protein-based methods, tumor-derived cfDNA are DNA fragments released from dying cancer cells, and DNA copy numbers are limited in a cell. Thus, there is a limit of detection and a potential limit to how early detection can occur. Thus, if a cancer-associated MAF is detected, it is likely cancer. Protein biomarkers are released by cancer cells at a relatively high amount, so are easily detectable early [39,66,67], but lack specificity because protein biomarkers can be released by both cancer cells as well as normal cells.

The cost of cfDNA testing has significantly decreased in recent years, although it is still over five times the price of protein biomarker panels [5]. However, it can generally be achieved at a cost below USD 1000. The price reduction may lead to increased accessibility; however, there are still some inherent issues with cfDNA testing that remain unresolved. One critical concern is its short half-life, potentially as brief as a few minutes to hours [68]. Such a short half-life would result in an unstable cfDNA quantity in the specimen. Additionally, specimen preservation would pose a challenge, as the cfDNA could degrade within a few hours of in vitro storage. In contrast, protein biomarkers have a half-life lasting several days or even weeks [69,70]. These inherent issues may be the reasons why the effectiveness of cfDNA testing in MCED is not as promising as initially anticipated. In fact, a study suggests that combining cfDNA with protein biomarker testing does not yield better cancer efficacy than using protein biomarkers alone [71]. Further optimization is required for the use of cfDNA testing in MCED. The diagnostic performance of the cfDNA biomarkers is summarized in Appendix A.

## 4. Synergizing AI Algorithms for Biomarker Analysis

### 4.1. Classical Machine-Learning Techniques in Biomarker Interpretation

Harnessing ML in interpreting clinical inputs for classification or prediction is becoming a mainstream application nowadays in the medical field. Several studies have indicated that ML algorithms analyzing clinical [72], genetic [13], or protein biomarker [16] results can provide diagnoses similar to or even better than those made by physicians. What is noteworthy is that ML algorithms demonstrate greater consistency in pattern recognition, reducing interindividual differences. In the medical domain, there exists a wide variety of ML algorithms, including logistic regression, decision trees, random forests, support vector machines, and more [73,74]. Despite differences in the underlying logic of these algorithms, their design aims to identify specific patterns and relationships between the data and the predicted targets.

In cancer screening or diagnosis studies, the effectiveness of ML algorithms was compared with physician interpretation of tests [16,75]. In these studies, human physicians used the reference range-based single-threshold method: predicting the probability of cancer occurrence within the next year if any test item exceeded the reference range. Conversely, if all test items fell within the reference ranges, the individuals were predicted not to be at risk of cancer. While this interpretation method is straightforward, the effectiveness of cancer screening is not as high as that achieved by machine-learning algorithms. This suggests that physicians may not be as sensitive to specific data patterns in laboratory test results as ML algorithms. The possible explanation is that ML algorithms detect the “face/pattern of a disease” rather than only a few test items.

While ML algorithms appear to generally outperform physicians in interpreting multiple test items, there does not seem to be a particular advantage among different ML algorithms for lab data-based classification problems in the medical field. Although in individual reports, various algorithms, like the support vector machine [75], random forest [76], and logistic regression [16], have been reported to outperform others. In a review study, it was also noted that other ML algorithms do not show a clear superiority over traditional logistic regression (also categorized as an ML algorithm) [77]. In fact, most MCED products still adopt logistic regression as the ML algorithm. Galleri, a cfDNA-based MCED AI, is composed of two logistic regression models, one for cancer detection and the other for predicting the tissue of origin [78]. Protein biomarker-based MCED products, such as OneTest [6] (20/20 GeneSystems) and CancerSEEK (Exact Sciences) [15], also revealed the utility of the classical ML algorithms. Overall, despite some ML algorithms seeming more prominent in these studies, their advantages are very limited. In fact, the nature of the laboratory data themselves determines whether such classification problems have good predictive performance. The data have already predetermined the predictive performance, and the choice of which ML algorithm to use does not play a significant role [79].

The reason why data predetermine the outcomes can be explained by the fact that the lab data-based AI models are based on lab data, and the lab tests typically have a good signal-to-noise ratio [77]. Moreover, these test items have undergone a series of rigorous validations from the development stage, and were implemented in clinical settings for years [80]. Thus, the lab test items fundamentally have a certain correlation with the predictive phenotypes or diseases. On top of that, tests like proteomic panels consisting of peptides and proteins would not suffice as biomarkers on their own; instead, acquiring an ML strategy for their interpretation renders good predictive performance [81]. On the basis that the data themselves are composed of such strong predictors, ML models built on either theoretical foundations can easily identify hidden patterns in the data. In summary, for medical AI models with lab data as the input, the importance of good data far outweighs the significance of the ML algorithm used.

### 4.2. Unveiling Deep Learning’s Potential in Biomarker Analysis

In recent years, deep-learning (DL) algorithms have achieved significant success in the field of computer vision. In the domain of medical imaging, DL algorithms are widely employed for the development of image-recognition models. Medical images, such as electrocardiograms, chest X-rays, and computed tomography scans, are particularly well-suited for the application of DL algorithms. In these areas, DL algorithms demonstrate excellent performance, often approaching the level of human experts [82]. One key distinction between DL algorithms and traditional ML algorithms lies in feature engineering. Typically, when dealing with high-dimensional data, traditional ML algorithms require the use of feature-extraction or feature-selection methods to reduce the data dimensions in order to improve the prediction accuracy. In contrast to traditional ML algorithms, DL does not necessitate upfront feature engineering [83]. Therefore, DL offers the convenience of not requiring these preprocessing steps over traditional ML and provides a distinct advantage in practice.

While DL algorithms have achieved significant success, it appears that they do not necessarily outperform conventional ML algorithms in the medical domain. For instance, the traditional ML–random forest method attained higher diagnostic performance than DL in ultrasound breast lesion classification [84]. In a study predicting postoperative patient conditions, DL algorithms did not demonstrate higher predictive capabilities compared to traditional logistic regression [85]. In another study predicting drug resistance based on mass spectrometry data, random forest or XGBoost algorithms exhibited higher predictive abilities than DL [80]. In the field of MCED, models using DL algorithms to analyze protein biomarker results did not show higher cancer prediction capabilities than traditional ML algorithms like logistic regression [35]. In certain data structures where the data themselves contain strong predictors, the need for feature engineering in DL algorithms is not as apparent as in traditional ML algorithms [80]. Thus, the performance comparison between DL and ML depends significantly on the data structure inherent to the specific application [84]. In situations where there is no advantage in predictive performance, the use of DL algorithms to analyze lab test results becomes debatable. Due to the complex computations within the model, DL algorithms require more processing time to generate classification or prediction results [80]. Beyond the longer processing time, DL algorithms also consume more energy compared to traditional ML algorithms to produce predictive outcomes [80]. In an era where AI algorithms are gradually becoming a part of daily life, energy-intensive methods pose a higher carbon footprint, eventually facing serious challenges. While there may not be a significant advantage in the predictive performance, certain DL algorithms can assist in addressing clinical challenges encountered in MCED in the real world. Taking the field of predicting cancer risk using protein biomarkers as an example, the test panel provided by each diagnostic institution may vary, with only partial overlap in the panels tested. Additionally, if the items tested for each case only partially overlap at different time points, comparing risk predictions becomes challenging. In this regard, long short-term memory networks, with their flexibility and tolerance for missing values, prove to be a suitable solution for addressing such clinical issues [35].

## 5. Training and Validation of AI Models for MCED

### 5.1. Impacts of Training Dataset: Case-Control, Retrospective Cohort, or Prospective Cohort?

AI technology is very promising for many applications in medical fields [11,72,86]. However, the robustness of the medical AI models is suboptimal when deployed into real-world settings [87]. The suboptimal robustness indicates that the medical AI models perform well in training and validation, but such models fail to perform in a real-world deployment. While the underlying mechanisms are many, for the MCED AI model, the most crucial factor for a suboptimal performance would result from the inadequate selection of training datasets. The training dataset types can be categorized into a case-control cohort, retrospective cohort, and prospective cohort (Table 1). The difference on the training dataset determines what the MCED AI models learn. Basically, MCED AI would perform well when the training datasets mimic the real-world settings. By contrast, diagnostic performance of the MCED AI models would drop if the training datasets are considerably different from the real-world settings. Typically, training by dataset with a case-control design is the most susceptible to failure when deployed in the real world, even if the training datasets are reasonably designed according to classical principles of ML models training. In training an ML model of binary classification, the training datasets include cases with a positive label and cases with a negative label. For MCED, the positive label indicates positive for cancer diagnosis while the negative label indicates cancer-negative (i.e., healthy cases). Cancer cases and healthy cases are the only learning materials for the ML algorithms. Appropriate datasets are the key to successful and useful MCED models. By contrast, inappropriate datasets would lead to disastrous deployment, even though the models perform well in the training processes.

The differences between a case-control study and a real-world cohort study for cancer screening would be as follows:**On cancer cases:** Specimens of the cancer cases in a case-control study are typically collected in more advanced stages than the specimens of a real-world cohort study. The reason for that is that the specimens of the cancer cases in a case-control study are collected when the diagnosis of cancer has been made, which is often associated with symptoms/signs that are caused by cancers. In this case, the cancers show their malignant behaviors, like space occupying or mass effect. By contrast, specimens of the cancer cases in a real-world cohort are collected long before cancer diagnosis or any symptom/sign. Such conditions are usually closer to the health checkup population in the real world. Theoretically, biomarkers in presymptomatic or asymptomatic cancer cases would be closer to those of healthy controls than in the symptomatic cancer cases.**On healthy cases:** The number of healthy control cases in a case-control study is usually up to several hundred given the fact that the ratio of cancer versus control ratio is set around 1:1–1:4 [15,17,34,49]. The relatively small number cannot represent the large diversity in the healthy control cases. As a result, there are fewer outlier cases. Fewer healthy outliers would simplify the classification problem (i.e., classify cancers versus healthy). AI models trained with fewer healthy outliers may therefore not have a classification threshold that can be used in the real world.

Figure 2 illustrates the probability distributions of the AI models trained by the case-control dataset or the real-world dataset. Consider a case-control study with a ratio of cancer to health cases = 100:100 and a real-world cohort study with a ratio of cancer to healthy cases = 100:10,000. In the training and validation steps, AI models in the case-control study can easily reach a good performance when the cutoff of the risk score is in the range 10.49–15.49 (Figure 2A). By contrast, the optimal cutoff would be largely converged to a smaller range or a number (e.g., 15 in the example, Figure 2B). As illustrated in the plots, the cutoff of AI risk scores in the case-control study are not optimized for use in the real world. Many models (or many cutoffs) can have good classification performance. However, once in the real world, the performance of these AI models will be significantly weakened or changed. For example, when an AI model with Cutoff.1 (i.e., 10.49) is used in a real-world setting, the specificity would largely decrease; when Cutoff.2 (i.e., 15.49) is used, the sensitivity would largely decrease. Of note, AI models with either Cutoff.1 or Cutoff.2 attain nearly 100% for sensitivity or specificity, but fail to deliver similar performance in real world.

The failure of AI models that were trained from case-control studies in a real-world setting can be explained by t-value theory. According to the t-value formula, the disparity of the means relative to the variances between the two populations (cancer group vs. healthy group) determines the significance of the difference between populations [88]. For a case-control study, a larger difference between the averages of the two populations indicates a greater statistical significance between the two populations and renders it easier to be classified. Moreover, MCED AIs that are trained on case-control studies will not recognize the smaller differences in marker values that will be seen in early diagnosis, because such asymptomatic/presymptomatic cases are missing in such datasets. Thus, the sensitivity of a case-control-trained AI model will be greatly lost in an asymptomatic early population. Furthermore, it may rely more on markers that dominate later in the cancer development process.

### 5.2. Cross-Validation vs. Independent Testing: Generalizability or Continual Monitoring Matters?

In the general development of AI models, validating the performance of the model is a crucial step. There are various methodologies for model validation, including k-fold cross-validation (KFCV), nested k-fold cross-validation (NKFCV), and independent testing. KFCV is typically used for initial internal validation, meaning it uses a single-source dataset while developing and validating the model. NKFCV is employed for smaller datasets, and it involves placing the model-tuning steps in a separate inner layer to avoid overfitting. Independent testing is commonly considered the fairest method in the machine-learning training and validation process.

For MCED AIs trained by real-world datasets, special data processing is necessary to cope with the extremely balanced data structure, in which the cancer versus noncancer ratio is around 1:100 [16]. To better train AI models, typically the number of cases with a positive label (i.e., cancer) and the number of cases with a negative label (i.e., noncancer) should be comparable. The oversampling of cancer cases or the undersampling of noncancer cases can be adopted to create a balanced dataset for model training (Figure 3A). The undersampling of noncancer cases would be a more acceptable method in the medical community because no artificial data are created. In the model training step, stratified sampling rather than simple random sampling is adopted to divide the data into the training dataset and the validation dataset (Figure 3B). With stratified sampling, the case number of the minor subgroup (i.e., cancer cases) is elevated so there will be sufficient cancer cases in the training dataset [89]. Of note, the cancer versus noncancer ratio should be kept the same as that of the original real-world cancer screening dataset in the validation dataset as well as in the independent dataset. It is only by keeping the original cancer versus noncancer ratio that the static metrics of AI models can be accurately estimated.

Independent testing can further be divided into using data from a second or even a third healthcare institution to validate the AI model trained on data from the first institution (Figure 3C). This approach ensures that the AI model trained by the first institution is not limited to its specific context, but can be applied more broadly. This characteristic is referred to as generalizability and is typically regarded as one of key characteristics for AI models [90]. However, should the use of medical AI be predicated on its generality? This issue has raised considerable interest in recent years. Medical behaviors are significantly influenced by the local socioeconomic status and healthcare insurance, coverage, as well as reimbursement. In places where healthcare costs are low and accessibility is high (such as in East Asia), people are accustomed to undergoing annual cancer screenings, even without any symptoms [16]. Conversely, in places where cancer screening is expensive and highly inconvenient, individuals might only opt for screening when experiencing significant symptoms or discomfort. Therefore, despite both being considered cancer screening databases, there is a substantial difference between them: categorized into presymptomatic/asymptomatic or symptomatic cases. These two groups exhibit significant differences in the progression of the disease, but current research has paid less attention to this aspect. This phenomenon creates a challenge for AI models for cancer screening, as obtaining similar results across datasets from different locations is difficult due to the inherent differences in these datasets.

Because the data collected from different places exhibit considerable heterogeneity, and there are significant variations in healthcare systems and reimbursement structures. Therefore, pursuing the generality of AI models for cancer screening would contribute minimally to increasing the robustness of local medical services. In contrast, time-wise management appears to be a more locally relevant strategy for AI-driven MCED (Figure 3C). Before the AI models provide services, it is essential to verify whether the predictive accuracy remains stable across different years. Additionally, offering services locally and continuously, recurrently monitoring the model’s performance over time, helps ensure its stability. This involves training the model with local data and testing it with local data, aiming to provide better local healthcare services. The primary focus of local healthcare should be to serve the local community effectively, and there seems to be little benefit in overly pursuing generalization. Although this approach deviates from the conventional emphasis on the generalization of AI models, it maximizes healthcare benefits.

In summary, data and validation methods predetermine the performance of MCED AI products. Using the RWD of cancer screening rather than case-control datasets would fit better to the purpose of cancer screening. The goal of model validation would lie on validating an AI model that will benefit the target population the most.

## 6. Challenges and Opportunities

### 6.1. Data Quality and Quantity: Navigating the Complex Landscape

While the performance of medical AIs depends largely on data, the characteristics of the data would serve as the cornerstone for the performance and applications of MCED AI models. There are still several key challenges ahead, and careful consideration is needed. One of the foremost challenges revolves around the inherent heterogeneity in datasets sourced from various medical institutions and laboratories. The reasons for such variations are manifold. Firstly, differences arise at the level of analytical measurements, where, even for the same biomarker, various laboratories and medical facilities may employ reagents or platforms from different suppliers. Studies indicate that variations would exist between assays for the same biomarker on different analytical kits, emphasizing the need for extensive and rigorous validation before clinical implementation [91]. Additionally, discrepancies in the biomarkers that are tested in the panel also pose computational challenges and interpretation issues for AI models. Taking the example of protein biomarkers, one MCED screening kit may assess seven protein targets [16], while another MCED AI examines twelve protein targets [49]. Yet another screening kit may include both protein and gene targets [15]. While some items overlap among these kits, the differences would complicate the interpretation and comparison. In this case, there is often an absence of comprehensive input data during the computation of AI models. To address the missing values, reliance on various data imputation methods becomes necessary. The methods for data imputation are diverse [92], but the imputed values generated through these methods may not represent real-world values. This imputation stage introduces some biases into the calculations. Moreover, regional or national heterogeneity also plays a significant role. Beyond factors related to diverse ethnicities, variations in health insurance reimbursement systems and healthcare cultures across different regions or countries can profoundly impact data collection [93,94].

Acquiring a training dataset that can represent the real world is crucial for an MCED AI model to be effectively used in cancer screening. Ideally, such locally relevant data should be continuously collected without selection from local routine medical practices. However, collecting such datasets poses significant challenges due to the scarcity of cancer cases compared to healthy cases [95]. An extremely imbalanced dataset is inevitable in the real-world setting (Figure 3A). The primary difficulty is the inadequate collection of a sufficient number of cancer cases, especially for rarer cancer types. Under the supervised learning training framework, insufficient positive cases will prevent the training of a trustworthy AI model. Therefore, collecting a substantial number of cases is absolutely critical. Unfortunately, in the current framework, patients or healthcare institutes lack sufficient incentives to contribute data because there are no additional rewards, and there are concerns about the privacy and security of personal information [96]. To address this dilemma, apart from some open-source medical datasets [15,97], there are emerging blockchain-based technologies that incentivize patients to upload medical data [98]. These technologies not only reward patients for contributing their data, but also create a cyclical reward system for the AI model’s profits, all while preserving individual privacy through blockchain technology.

### 6.2. Interpretability, Explainability, and Integration: Bridging the Gap in AI-Driven Insights

In addition to the issues related to data and AI algorithms, the most significant challenge in implementing an MCED AI model in clinical applications is likely how to interpret and communicate the results generated by the AI models (Figure 4). With extensive research in recent years, AI models in healthcare have made significant progress. However, beyond predictive performance, correctly explaining the reasons for such predictions is crucial to gaining the trust of clinical physicians and patients. Information such as the coefficient in logistic regression or RF importance in random forests can provide rationale for predictions. Only when clinical physicians can clearly explain the reasons for the model’s interpretation can they trust that the AI model’s interpretation is reasonable and not just a “black box”. Establishing such a trust relationship between clinical physicians and AI models is a crucial first step for the success of MCED AIs. Moreover, clinical factors like age or sex are informative and crucial inputs for clinical diagnosis. The importance of such clinical factors would be as important as the serum biomarkers [16]. While these clinical factors are the typical factors to be considered in the clinical diagnosis, including more clinical factors together with the serum biomarkers in MCED AI models would provide a more comprehensive solution to healthcare professionals.

On top of a reasonable and explainable MCDE AI model, the next crucial question that physicians would be interested in is the clinical relevance and actionability [99]. To communicate with patients, the MCED AIs should provide more communicable terms for the predictive results. The predictive probability generated by AI models may not be an adequate metric for communication with patients. Instead, the incidence-based risk score (or positive predictive score, PPV score) that has been widely used in prenatal checkups is a more appropriate term [100]. The PPV score is based on comparing the patient’s predictive probability to the cases with similar risk levels. For example, a risk of 1 in 10 is interpreted as higher than the background risk, whose risk level is 1 in 1000. The population-derived PPV score would be more intuitive for nonmedical professionals. Additionally, the PPV score is also an explicit metric rather than simply a predictive probability to have in comparison to the background risk.

Unfortunately, only a few MCED AI models provide actionability to clinical physicians and patients, leaving a considerable gap between a report of elevated risk and the following cancer diagnosis. For cases with an elevated risk for cancers, the next diagnosis of interest is the staging and localization. Staging relevant to the risk score would provide earlier information for prompt action [6]. Localization is also key information for MCED AI models. When the tissue of origin is provided together with the risk score [6,15,18], physicians would know which medical specialty to suggest to the patients with an elevated risk score. Actionable suggestions following an MCED test should also be provided to complete the whole test-and-action cycle. The call-to-actions that are built based on evidence can include the time interval for following-up, retesting, and visiting medical subspecialties [6,18]. All of the actions can be depicted in a flowchart for easy guidance for MCED-using physicians so that an MCED AI can be seamlessly integrated into clinical workflows, facilitating informed decision making by healthcare professionals.

## 7. Conclusions

Serum biomarker-based AIs hold promise in MCED and are undergoing rapid development. The analytical targets include cfDNA, protein biomarkers, or their combination. When the serum biomarkers typically have the characteristics of strong predictors, various ML algorithms can have good diagnostic performance. The technical key of building a trustworthy MCED AI resides in using real-world mimetic data rather than a case-control design for training and validation, so as to have a robust implementation back in a real-world setting. Like other medical products, MCED AIs with high interpretability, explainability, and actionability would integrate better into medical workflows and benefit more patients in early cancer diagnosis.

## Figures and Tables

**Figure 1 cancers-16-00862-f001:**
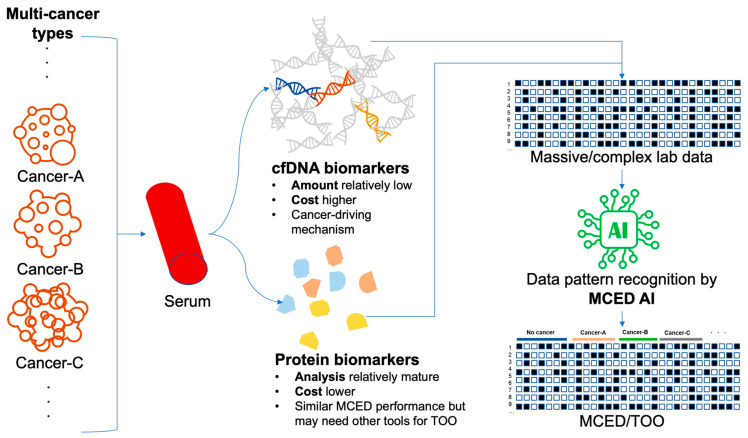
Illustrative scheme of MCED, serum biomarkers, and MCED AI.

**Figure 2 cancers-16-00862-f002:**
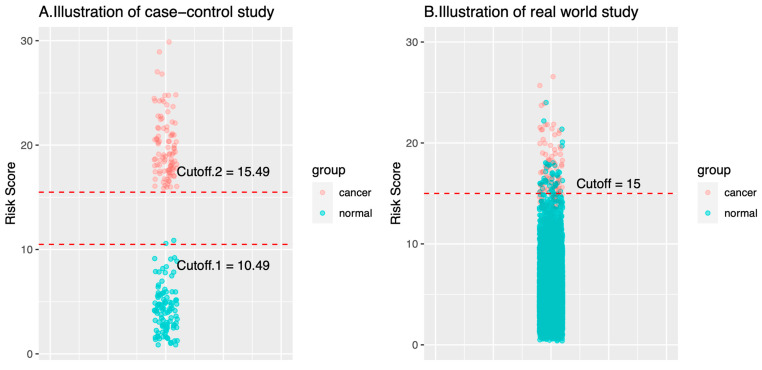
Probability distribution difference between cancer cases and normal (healthy) cases in (**A**) the case-control study dataset and (**B**) the real-world cancer screening dataset. In a case-control study dataset, cancer cases and noncancer cases are well-defined at the time of enrollment. The risk score distributions of cancer and noncancer cases would be apparently different. In this case, any cutoff value in between Cutoff.1 and Cutoff.2 is fine to have a perfect predictive performance. By contrast, the risk score distributions of cancer and noncancer cases in the real-world cancer screening dataset overlap more, and the optimal diagnostic cutoff is much more narrow than those in the case-control study. The illustrative plots demonstrate the reason why the MCED AI models that are trained by using the data of case-control studies would have suboptimal predictive performance in a real-world cancer screening.

**Figure 3 cancers-16-00862-f003:**
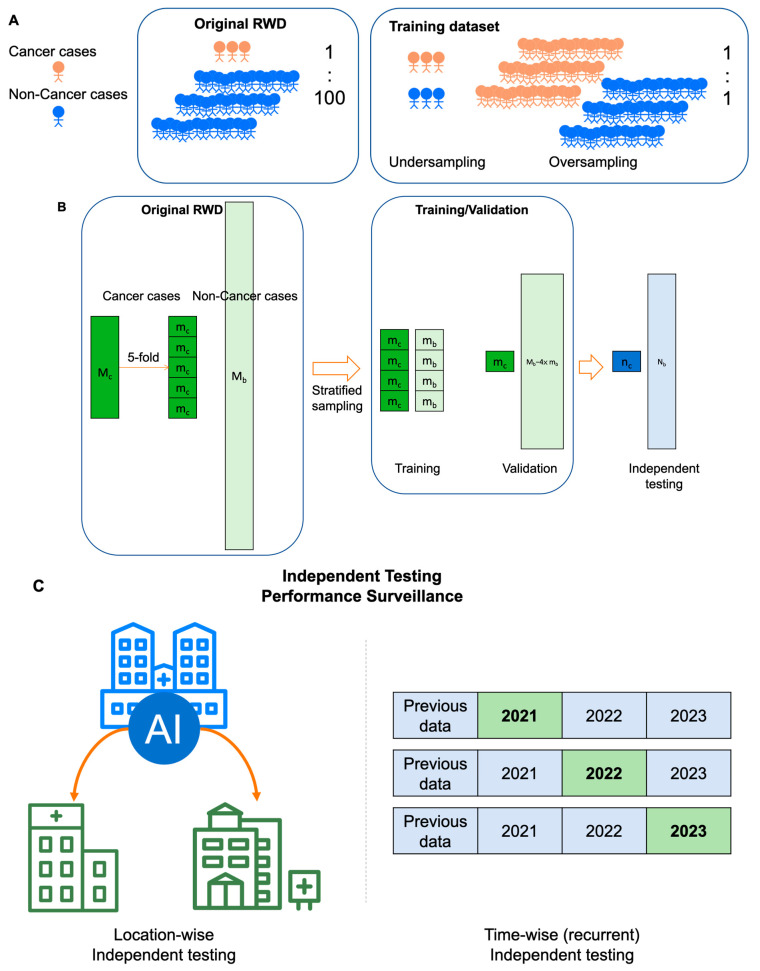
Special considerations on developing MCED AI models. (**A**) The paucity of cancer cases in the real-world cancer screening scenario. In a real-world cancer screening dataset, the ratio of cancer cases versus noncancer cases is typically around 1:100. The ratio is extremely unbalanced for training an AI model. The oversampling of cancer cases or the undersampling of noncancer cases are commonly used data processing methods to create a balanced dataset for AI model training. (**B**) Data processing for training, validating, and independently testing MCED AI models. In the undersampling strategy, stratified sampling can be used to create a balanced training dataset. By contrast, the cancer versus noncancer cases ratio is good to be kept the same as the original dataset for both validation and independent testing in order to have an accurate estimation of diagnostic metrics. M_c_: cancer cases; m_c_: cancer cases in 5-fold split datasets; M_b_: noncancer cases; m_b_: noncancer cases in 5-fold split datasets that are sampled from M_b_ by using stratified random sampling; N_c_ and N_b_: cancer cases and noncancer cases in an independent dataset. (**C**) Different approaches for the independent testing of MCED AI models. Location-wise independent testing can be used to test the generalizability of an AI model across different locations. Time-wise independent testing can be used to recurrently test the robustness of an AI model in different periods of time.

**Figure 4 cancers-16-00862-f004:**
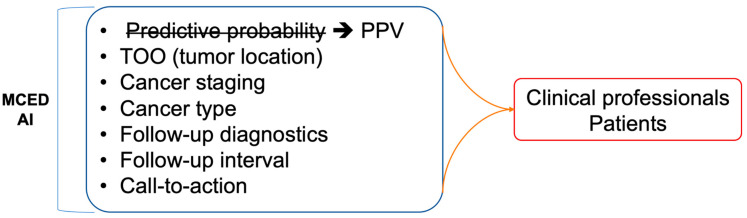
Challenges for the implementation of MCED AIs. Besides cancer early detection, an MCED AI product should also provide a lot of clinically relevant information in order to successfully integrate MCED AIs into current clinical workflows of diagnosing and treating cancers.

**Table 1 cancers-16-00862-t001:** Serum biomarker-based MCED AI products on the market. RWD: real-world dataset; CCD: case-control dataset; Sen: sensitivity; Spe: specificity; NNS: number needed to screen; TOO: tissue of origin.

MCED Products	Biomarkers	Cancer Types	Algorithms	Model Development	Performance	Report	Comments
**Gallery [18,78]**	cfDNA methylation (>100,000 informative methylation regions)	More than 50 types	Logistic regression	**Train:** CCD **Validation:** CCD-based independent testing **Prospective** RWD	Sen: 28.9% Spe: 99.1% NNS: 189	cancer detected/not detected TOO	Validated in a prospective cohort study. Prediction of TOO is accurate. Market available. Cost is relatively high (USD 949) but may decrease with DNA sequencing cost in the future. Top 3 cancer types with the best performance: head and neck, pancreas, and lymphoma; suboptimal performance in kidney, prostate, and breast cancers.
**OneTest [6,35]**	Protein biomarkers (six tumor markers for male: AFP, CEA, CA19-9, CYFRA21-1, SCC, and PSA, and seven protein tumor markers for females: AFP, CEA, CA19-9, CYFRA21-1, SCC, CA125, and CA15-3)	More than 20 types	Classical ML algorithms; Long short-term memory algorithm	**Train:** RWD **Validation:** RWD-based cross-validation RWD location-wise independent testing	Sen: 82.3% Spe: 80.8% NNS: 125 (male); 200 (female)	cancer detected/not detected categorized risk TOO	Developed and validated by using RWD. Market available with affordable cost (USD 189), potential for large scale MCED. Top 3 cancer types with the best performance: prostate, colon, and liver; suboptimal performance in breast cancers, cervical cancers, and lymphoma.
**OncoSeek [17]**	Protein biomarkers (seven protein tumor markers: AFP, CA125, CA15-3, CA19-9, CA72-4, CEA, and CYFRA 21-1)	Nine types: breast, colorectum, liver, lung, lymphoma, osophagus, ovary, pancreas, and stomach	Classical ML algorithms	**Train:** CCD **Validation:** CCD-based cross-validation CCD-based independent testing	Sen: 51.7% Spe: 92.9%	cancer detected/not detected TOO	Cost-effective MCED tool. Retrospective RWD or prospective cohort study needed for validation. Top 3 cancer types with the best performance: pancreas, ovary, and liver; suboptimal performance in breast cancers, esophagus cancers, and lymphoma.
**CancerSeek [15]**	cfDNA + protein biomarkers (61 genetic markers and 8 protein tumor markers: CA125, CA19-9, CEA, HGF, myeloperoxidase, OPN, prolactin, TIMP-1)	Eight types: ovary, liver, stomach, pancreas, esophagus, colorectum, lung, and breast	Logistic regression	**Train:** CCD **Validation:** CCD-based cross-validation	Sen: 70% Spe: 99%	cancer detected/not detected TOO	Partial markers (protein biomarkers alone) would perform as well as the full panel. Top 3 cancer types with the best performance: ovary, liver, and stomach; suboptimal performance in breast, lung, and colorectum cancers.

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
