# Peer review of "Integrating Artificial Intelligence for Advancing Multiple-Cancer Early Detection via Serum Biomarkers: A Narrative Review"

_cancers, 2024, doi:10.3390/cancers16050862_

Round 1

Reviewer 1 Report

Comments and Suggestions for Authors

The authors review various AI-based MCED products, analyzing their components, training/validation approaches, and clinical applications. The work is clear and I would only suggest a couple of minor points to be addressed:

1.The paper is well structured. However, it would benefit from a more concise introduction of the inadequacy of current screening strategies.

2.The sections introducing MCED models should be more detailed, particularly in explaining how AI algorithms specifically enhance MCED.

3. A more critical analysis comparing these products could provide deeper insights. This comparison could highlight strengths, weaknesses, and potential improvements.

Comments on the Quality of English Language

The paper would benefit from a more efficient summarization of the main points, especially in the introduction.

Author Response

Dear Reviewer:

Thank you for the constructive comments and detailed reviewing of our manuscript. We have addressed your comments or concerns in the revised manuscript. Please refer to the attached response letter for the details.

Best regards,

HsinYao Wang

Reviewer 2 Report

Comments and Suggestions for Authors

In recent years, the concept and policy of multiple cancer early detection (MCED) has received significant attention from governments around the world. This review examines several mature MCED AI products currently available in the market, identifying their constituent factors based on serum biomarker detection, MCED AI training/validation, and clinical application. The review highlights the challenges that existing MCED AI products face during these phases and offers insight into the ongoing developments and obstacles in the MCED AI field.

1. Authors should add summary tables to sections 3.1, 3.2, and for all parameters/parameter combinations they would like to see sensitivity and specificity values confirmed in clinical studies.

2. In section 4, I would like to include a summary table of the algorithms used in MCED AI products with a conclusion about which of them are the most promising.

3. According to Figure 1: it may make sense to introduce the concept of a “gray zone”, when we have two threshold values that allow us to obtain high values of sensitivity and specificity, and all values between them will fall into the “gray zone”, so additional ones are needed to make a diagnosis .

4. Table 2 and Figure 1 are of very poor quality and need to be corrected.

5. Table 1 needs to be improved; it is necessary to indicate which biomarkers are included in the test, for which types of cancer this test system works well, for which it does not, and provide sensitivity and specificity values.

Author Response

(The authors gave the same response as above.)

Reviewer 3 Report

Comments and Suggestions for Authors

Dear authors,

Congratulations for your great work. Indeed you have approached a novel and rapid developing approach to cancer detection that is very interesting, minimal invasive, rather cheap and holds potentially great clinical value. 

I have no comments except check your authors: "Michael
Lebowitz and PhD" is probably MD and PhD and your abstract: "The concept and policies of multiple early cancer detection (MCED)", the concept most probably being Multi-Cancer Early Detection as very correctly stated elsewhere in the paper.

Author Response

Dear Reviewer:

Thank you for the detailed reviewing of our manuscript. We have addressed your comments and revised the manuscript. Please refer to the attached response letter for the details.

Best regards,

HsinYao Wang

Reviewer 4 Report

Comments and Suggestions for Authors

The manuscript discusses the impacts of training datasets on the performance of AI models for Medical Cancer Early Detection (MCED). It emphasizes the importance of real-world datasets over case-control designs and highlights the challenges of data quality, quantity, and model interpretability. The reviewer suggests clarifying the definition of "MCED AI," including figures in the text, and providing a more detailed explanation of the discussed probability distributions. Overall, the manuscript holds promise but requires improvements in clarity and visual representation. While generally well-structured and interesting to read some comments for authors consideration are listed below:

Comments for authors:

Title:

The title accurately reflects the content of the study and is concise and informative.

Review body:

  1. Introduction of AI Section: The transition from discussing the challenges in cancer screening to the role of AI in MCED is somewhat abrupt. Consider providing a brief statement or sentence that explicitly connects the two sections, ensuring a smoother transition.
  2. Details on AI Applications: While the section mentions various applications of AI in the biomedical field, a bit more detail on specific AI applications related to cancer early detection could enhance the reader's understanding. For example, elaborating on how AI is applied in liquid biopsy or genetic data analysis would add depth to the discussion.
  3. Conclusion of the Section: The section lacks a clear conclusion summarizing the key points discussed. Including a concluding paragraph would help wrap up the ideas presented and prepare the reader for the next section.
  4. Acronyms and Abbreviations: While the use of acronyms and abbreviations is generally appropriate, consider including a list of abbreviations or acronyms at the beginning of the document to ensure clarity, especially for readers who may not be familiar with the terminology.
  5. Transition to cfDNA Section: The transition from discussing protein biomarkers to cfDNA could be made more explicit. Consider adding a sentence that smoothly connects the two subsections, providing a clearer link for the reader.
  6. Visualization: The section includes complex technical details, and the inclusion of figures or diagrams could aid in visualizing concepts like the detection limits of cfDNA or the structure of protein biomarker panels. Visual aids can enhance reader understanding.
  7. Practical Examples: Providing specific examples or case studies where ML or DL algorithms have been successfully applied in biomarker analysis could make the content more tangible for readers.
  8. A concise concluding paragraph summarizing the key points about the impacts of training datasets and the importance of model validation for MCED could reinforce the main ideas.
  9. Figure references in the text (e.g., "Figure 1A," "Figure 2B") are mentioned, but the figures are not provided in this submission. Ensure that the figures are included for a comprehensive review.

Author Response

(The authors gave the same response as above.)

Round 2

Reviewer 2 Report

Comments and Suggestions for Authors

I have no further comments/remarks on the article. I believe that in its present form the manuscript can be recommended for publication.